# Characterization and Transcriptome Analysis Reveal Abnormal Pollen Germination in Cytoplasmic Male Sterile Tomato

**DOI:** 10.3390/ijms26178337

**Published:** 2025-08-28

**Authors:** Kosuke Kuwabara, Tohru Ariizumi

**Affiliations:** 1Graduate School of Agricultural Science, Tohoku University, Sendai 980-8572, Miyagi, Japan; 2Graduate School of Life and Environmental Sciences, University of Tsukuba, Tsukuba 305-8572, Ibaraki, Japan; 3Institute of Life and Environmental Sciences, University of Tsukuba, Tsukuba 305-8572, Ibaraki, Japan; 4Tsukuba Plant Innovation Research Center, Tsukuba 305-8572, Ibaraki, Japan

**Keywords:** tomato, cytoplasmic male sterility, transcriptome, pollen germination, pectin methylesterase inhibitor

## Abstract

Cytoplasmic male sterility (CMS) is a plant trait wherein plants cannot develop normal male organs because of the mitochondrial genes. Although the mitochondrial gene *orf137* has been identified as the CMS-causing gene in tomatoes, its function remains unclear. In this study, we characterized the sterile male phenotypes and analyzed the CMS pollen transcriptome. Microscopic and calcium imaging analyses revealed that CMS pollen exhibited abnormal germination from multiple apertures, accompanied by elevated calcium concentrations and vesicle accumulation, which are typically observed in pollen tube tips. RNA-Seq analysis revealed 440 differentially expressed genes, including four *pectin methylesterase inhibitor* (*PMEI*) genes that were highly expressed in the pollen. PME activity was significantly reduced in CMS pollen, suggesting its association with abnormal pollen germination. ATP and reactive oxygen species (ROS) levels, which are key mediators of mitochondrial retrograde signaling (MRS), remained unchanged in CMS pollen, and the expression of the mitochondrial stress marker *AOX1a* was not elevated. These findings suggest that *orf137* triggers an alternative MRS pathway independent of ATP or ROS, potentially leading to *PMEI* upregulation and abnormal pollen germination. Our results reveal a previously unrecognized mechanism of CMS-induced male sterility in tomatoes involving nuclear gene regulation through unconventional mitochondrial signaling.

## 1. Introduction

Cytoplasmic male sterility (CMS) is a plant trait wherein plants cannot develop normal male organs because of the mitochondrial genes, called CMS-causing genes. CMS has been identified in more than 150 species of higher plants [1]. CMS plants cannot produce seeds through self-pollination; therefore, they are widely used in hybrid breeding systems because they eliminate manual emasculation and ensure high hybrid seed purity.

CMS-causing genes have been identified across various plant species, and several models have been proposed to elucidate their mechanisms of action. Currently, four major models are widely discussed: (1) energy deficiency, (2) aberrant programmed cell death (PCD), (3) cytotoxicity, and (4) retrograde regulation models [2,3]. In the energy deficiency model, most CMS-causing gene products contain one or more transmembrane domains that can disrupt mitochondrial respiration by interfering with the components of the electron transport chain. This may reduce ATP synthesis and overproduction of reactive oxygen species (ROS). Since male reproductive development is highly dependent on ATP, a decline in mitochondrial ATP production is conjectured to directly disrupt the male reproductive organs [2,4]. The aberrant PCD model proposes that the tapetum of the anther undergoes precisely regulated PCD at specific developmental stages through a process mediated, in part, by ROS. The tapetum is crucial for supplying proteins, lipids, and other components essential for pollen development. However, excessive ROS production by CMS-causing proteins can trigger premature tapetum degeneration, leading to male sterility [5]. The cytotoxicity model has shown that CMS-causing proteins exhibit cytotoxicity. For example, certain CMS proteins localize to the inner mitochondrial membrane, forming polymers or channels that cause electrolyte leakage and ultimately exert cytotoxic effects. These effects have been observed in *Escherichia coli* and yeast overexpressing CMS-causing proteins [4]. The retrograde regulation model hypothesizes that CMS-causing genes influence the expression of nuclear genes through mitochondrial retrograde signaling (MRS). Direct evidence of MRS involvement in CMS has been reported in Chinese wild (CW)-type rice. In this system, the *Rf17* gene, which encodes a protein with an acyl carrier protein synthase-like domain, restores fertility when a specific nucleotide substitution is present in the promoter region. In the cytoplasmic background of CMS, this substitution reduces *Rf17* expression, paradoxically leading to fertility restoration [6,7]. Recently, retrograde regulation has received increasing attention as a potential mechanism of CMS. Transcriptome analyses are commonly used to comprehensively identify nuclear genes whose expression is altered in the cytoplasmic background of CMS, with the goal of elucidating the downstream targets of mitochondrial signaling that contribute to male sterility.

Recently, our group identified *orf137* as a mitochondrial gene in tomato CMS lines but not in fertile cultivars [8]. This gene shares mild sequence similarity with *orf507*, a known CMS-causing gene in pepper [9]. Functional analysis using mitoTALEN-mediated knockout of *orf137* fully restored fertility in tomato CMS lines, providing strong evidence that *orf137* is the causal gene responsible for CMS in tomatoes [10]. However, the molecular mechanism of *orf137-*induced male sterility remains unclear. Therefore, we first investigated the phenotypic characteristics of male sterility in the tomato CMS line. We focused on detailed observations of vesicle and Ca^2+^ localization, both of which are critical for pollen germination, to better understand the abnormalities in this process. Additionally, we conducted transcriptome analysis of pollen to identify nuclear genes whose expression was altered in the CMS line and may be associated with male sterility. Based on these findings, we discuss the potential mechanistic relationship between *orf137*, changes in nuclear gene expression, and the manifestation of male sterility in tomatoes.

## 2. Results

### 2.1. Abnormal Phenotypes During the Pollen Germination Stage in the Tomato CMS Line

First, the phenotypes of mature pollen in tomato CMS line CMS[P] were compared with the maintainer line P. Pollen grains from both lines were stained with I_2_-KI solution, indicating the accumulation of starch (Appendix A). Subsequent 4′,6-diamidino-2-phenylindole (DAPI) staining further revealed the presence of both vegetative and generative nucleus in the pollen of both lines (Appendix A). In addition, transmission electron microscopy (TEM) analysis of mature pollen showed that CMS[P] pollen contained such organelles as starch granules, vacuoles, and mitochondria, along with intine and exine. These cellular components and structures were comparable to those observed in maintainer line P (Appendix A). We previously developed the tomato CMS line Dwarf CMS[P] by backcrossing the CMS[P] with the dwarf tomato cultivar Micro-Tom. This line retains the same mitochondrial genome as CMS[P], including the CMS-causing gene *orf137* [10]. The morphology of the plant was identical to that of Micro-Tom (Figure 1A). There were no differences in flower structures between the two lines (Figure 1B). To investigate the pollen viability, pollen was stained by DAPI and I_2_-KI solution. Mature pollen grains of both Micro-Tom and Dwarf CMS[P] exhibited normal staining of the vegetative and generative nuclei (Figure 1C). Additionally, pollen from both lines was stained by I_2_-KI solution, with comparable staining rates observed (Figure 1D,E). These results indicate that pollen staining assays are insufficient to determine pollen sterility in Dwarf CMS[P].

Next, the germination capability of pollen from Micro-Tom and Dwarf CMS[P] was compared. Pollen from Micro-Tom successfully germinated on both the artificial medium and the stigma. Conversely, pollen from Dwarf CMS[P] exhibited abnormal germination phenotypes, characterized by swelling or bursting at two or three apertures (pollen tube emergence sites) (Figure 2A,B). We quantified the proportion of pollen exhibiting these phenotypes at 2, 4, 6, and 24 h after incubation. In the Micro-Tom, approximately 90% of the pollen germinated within 2 h. In contrast, Dwarf CMS[P] pollen did not germinate (0% germination) within 2 to 24 h. Micro-Tom displayed approximately 0% multiple swollen or burst apertures throughout the incubation period. In contrast, Dwarf CMS[P] exhibited 34% of the pollen with multiple swollen apertures at 2 h, which gradually decreased as incubation continued. The percentage of pollen grains with burst apertures increased from 9% at 2 h to 88% at 24 h (Figure 2C). These findings indicate that pollen from the tomato CMS line undergoes abnormal morphological changes during germination induction, particularly swelling and bursting at multiple apertures.

### 2.2. Pollen Germination at Multiple Sites in Tomato CMS Line

The internal structure of pollen from Dwarf CMS[P] was investigated using TEM to explore its unusual pollen morphology before and after incubation in liquid medium. In pollen before incubation, organelles as well as intine and exine layers were observed in both lines (Figure 3A1–A3,C1–C3). After incubation of the Micro-Tom pollen, vesicles were predominantly localized at the tip, whereas mitochondria and other organelles were absent (Figure 3B1–B3). This observation was consistent across the six independent pollen samples. Similarly, numerous vesicles were observed in the swelling apertures of Dwarf CMS[P] pollen (Figure 3D1–D3), with consistent results across all 11 independent pollens.

A cytosolic Ca^2+^ gradient is formed during pollen germination following hydration, and Ca^2+^ concentration ([Ca^2+^]_cyt_) increases at the site where the pollen tube emerges. Furthermore, a persistent Ca^2+^ gradient is maintained at the tip of the elongating pollen tube throughout its growth [11,12]. To visualize Ca^2+^ dynamics during pollen germination and pollen tube elongation, we used the Ca^2+^ indicator G-CaMP5 [12], which is specifically expressed in pollen under the control of the *Lat52* promoter. In Micro-Tom pollen, [Ca^2+^]_cyt_ initially increased at multiple germination apertures, but just before germination, a strong [Ca^2+^]_cyt_ signal was concentrated at a single aperture from which the pollen tube emerged (Figure 4A). After germination, a strong [Ca^2+^]_cyt_ signal was consistently maintained at the tips of the elongated pollen tubes (Figure 4B). In Dwarf CMS[P] pollen, multiple apertures also showed strong [Ca^2+^]_cyt_ signals. Swelling of the germination apertures was observed, along with intense [Ca^2+^]_cyt_ accumulation, followed by similar swelling events in other apertures. Strong [Ca^2+^]_cyt_ signals persisted at the tips of the swollen apertures (Figure 4C). These results suggest that, rather than merely swelling after incubation, Dwarf CMS[P] pollen attempts germination from multiple apertures, with [Ca^2+^]_cyt_ dynamics similar to those observed during normal pollen tube formation.

### 2.3. Transcriptome Analysis in Pollen of Tomato CMS Line

Micro-Tom pollen begins to germinate after 15 min of incubation in a pollen germination medium (Appendix A), and transcriptomic differences due to morphological changes may arise beyond this time point. To minimize the spatial and structural effects on gene expression, and thereby directly identify genes regulating pollen germination, RNA-Seq was performed using pollen samples incubated for 10 min. As a result, a total of 440 differentially expressed genes (DEGs) were identified between Micro-Tom and Dwarf CMS[P] (|log_2_ fold change (FC)| > 1, false discovery rate (FDR) < 0.1), of which 177 were upregulated and 263 were downregulated in Dwarf CMS[P] compared to Micro-Tom (Figure 5, Appendix A).

Among the DEGs, we focused on genes related to pectin modification because the pollen germination phenotype observed in Dwarf CMS[P] closely resembled that of the Arabidopsis *pme48* mutant, which exhibits pollen tube emergence or swelling from multiple apertures (or called “double-tipped”) [13]. Pectin methylesterase (PME) and PME inhibitor (PMEI) are key regulators of pectin methylation and have been reported to affect pollen tube formation [14]. To comprehensively identify *PME* and *PMEI* genes in tomatoes, we performed HMMER searches [15] using the Pfam domain profiles, Pfam01095 and Pfam04043, which correspond to *PME* and *PMEI*, respectively. Consequently, 77 *PME* and 48 *PMEI* genes were identified in the tomato nuclear genome. While none of the 77 *PME* genes showed significant expression changes, four of the 48 *PMEI* genes (*Solyc02g069300*, *Solyc05g005030*, *Solyc09g092350*, and *Solyc10g081670*) were significantly upregulated in Dwarf CMS[P] (Appendix A). Subsequently, the tissue-specific expression patterns of all 48 *PMEI* genes were investigated using z-score normalization. Among them, five *PMEI* genes, including four identified as DEGs, were expressed in the male reproductive organs (Figure 6A). When examining the actual transcripts per million (TPM) values, the four *PMEI* DEGs were expressed at higher levels in incubated pollen than in the mature anthers. In contrast, *Solyc01g059940*, which was not identified as a DEG, exhibited similar expression levels between mature anthers and incubated pollen and the lowest expression among the five DEGs in the incubated pollen (Figure 6B). These findings suggest that the four *PMEI* DEGs are preferentially and highly expressed in pollen among the 48 *PMEI* genes and are likely responsible for regulating pectin methylesterification in pollen.

To further assess the functional relevance of *PMEI* upregulation, PME enzyme activity was measured using total protein extracted from pollen incubated for 10 min. Consistent with the transcriptomic findings, PME activity in Dwarf CMS[P] pollen was significantly reduced to approximately half the level observed in Micro-Tom (Figure 6C). This reduction in PME activity indicates that the upregulated *PMEI* genes in Dwarf CMS[P] actively inhibit PME function. Overall, these results suggest that the elevated expression of specific *PMEI* genes and the consequent suppression of PME activity may contribute to the abnormal pollen germination phenotype characteristic of Dwarf CMS[P], particularly the emergence from multiple apertures.

### 2.4. Assessment of ATP and ROS Production in Tomato CMS Pollen

All four *PMEI* genes that were strongly expressed in pollen were upregulated in Dwarf CMS[P], suggesting the existence of a regulatory mechanism in which the mitochondrial CMS-causing gene *orf137* influences the expression of these nuclear genes via MRS. ATP and ROS are well-known inducers of MRS [16]. In other plant species, CMS-causing genes induce a loss of function in the respiratory chain complex, resulting in decreased ATP and increased ROS levels in male organs [2,4]. Therefore, we investigated whether ATP and ROS production fluctuated in the pollen of tomato CMS lines. Fluorescent labeling of ROS within the pollen using H_2_DCFDA revealed comparable fluorescence and intensities in the pollen of Micro-Tom and Dwarf CMS [P] (Figure 7A,B). In addition, ATP content was similar in the two lines (Figure 7C). These results suggest that pollen from tomato CMS lines maintains normal ATP and ROS levels. *AOX1a* is commonly used as a marker gene for mitochondrial stress responsiveness because its expression increases during stress [16,17]. RNA-Seq data showed no variation in *AOX1a* expression levels between Micro-Tom and Dwarf CMS[P] pollen (Figure 7D). These observations imply that ATP production and ROS development are normal in Dwarf CMS[P] pollen and that there are distinct signaling pathways from conventional mitochondrial stress that regulate the four nuclear-encoded *PMEI* genes.

## 3. Discussion

### 3.1. Tomato CMS Lines Exhibit a Unique Male Sterility Phenotype

In a previous study, we identified *orf137* as the gene responsible for CMS in tomatoes [10]. In this study, we investigated the mechanism by which *orf137* induces male sterility in tomatoes. The CMS line showed no morphological abnormalities in vegetative or floral organs, and pollen staining revealed no apparent differences from the maintainer line, suggesting normal pollen development. Unlike typical CMS lines in other plant species, which often display anther deformation, pollen abortion, or unstainable pollen [2,4], the tomato CMS line exhibits a markedly different phenotype.

TEM revealed concentrated vesicles at the tip of the pollen tubes in Micro-Tom. In contrast, mitochondria and other organelles were not observed at the tips of the tubes. This phenomenon is well documented in Arabidopsis pollen, where vesicles are actively transported to the pollen tube tip along actin filaments by myosin motors. In contrast, larger organelles, such as Golgi bodies and mitochondria, move along actin filaments in the shank region of the pollen tube rather than accumulating at the tip. Since vesicles contain cell wall materials, proteins, and other components, their accumulation at the pollen tube tip and subsequent membrane fusion are critical for pollen tube elongation [18,19]. Tip-localized vesicles were also observed at the swollen apertures of CMS pollen. Thus, it is suggested that tomato CMS pollen retains the intracellular organelle transport mechanisms necessary for pollen tube growth, similar to fertile pollen. In addition, it is well known that [Ca^2+^]_cyt_ accumulates at the site where the pollen tube emerges during germination, and a consistent Ca^2+^ gradient is maintained at the tip of the pollen tube [11,12]. This tip-focused Ca^2+^ gradient is important for controlling cell growth, vesicular transport, and intracellular signaling [20]. These phenomena were observed in the pollen tubes of Micro-Tom and the swollen apertures of Dwarf CMS[P] pollen. Overall, TEM and Ca^2+^ imaging results indicated that phenomena consistent with pollen germination occurred at multiple apertures in tomato CMS pollen grains. These findings suggest that tomato CMS pollen exhibits a unique phenotype characterized by germination from multiple apertures.

### 3.2. Upregulated PMEI Gene Expression Is Associated with Male Sterility in the Tomato CMS Lines

PME catalyzes the demethylesterification of homogalacturonan, which is a major pectin component. PME activity is precisely regulated by PMEI through direct protein–protein interactions. The balance between PME and PMEI determines the degree of pectin methylesterification, which affects the mechanical properties of the cell walls. Mutations in *PME* or *PMEI* often result in various phenotypic changes, including altered plant growth, responses to abiotic stress, and defects in pollen development [21]. In Arabidopsis, the *pme48* mutant shows reduced PME activity in pollen and a higher frequency of pollen grains exhibiting swelling or emergence of pollen tube-like structures from two of the three apertures, a phenotype not typically observed in wild-type plants [13]. In the tomato CMS lines examined in this study, no significant changes in *PME* gene expression were observed. However, four *PMEI* genes that were highly expressed in mature pollen were upregulated. A reduction in PME enzymatic activity was consistently observed in the pollen of the CMS line, likely due to the increased expression of *PMEI* genes. Considering the relationship between reduced PME activity and abnormal pollen germination observed in the Arabidopsis *pme48* mutant [13], the decreased PME activity detected in tomato CMS pollen was conjectured to be associated with the emergence of pollen tubes from multiple apertures. However, the mechanism by which reduced PME activity results in pollen germination from multiple apertures remains unclear and requires further investigation.

### 3.3. Normal ATP and ROS Production in Tomato CMS Line

Most CMS-causing genes encode proteins with one or more transmembrane domains and interfere with mitochondrial respiration by disrupting components of the electron transport chain. This disruption can result in decreased ATP production and elevated ROS levels [2,3,4]. The tomato CMS-causing gene *orf137* encodes a protein predicted to have a single transmembrane domain [8] and was initially hypothesized to reduce ATP synthesis and elevate ROS levels in the pollen. However, contrary to this hypothesis, ATP and ROS levels were comparable between the CMS and maintainer pollen. Alterations in mitochondrial ATP and ROS levels are recognized as triggers for MRS, a process in which the functional state of mitochondria regulates the expression of nuclear genes [16]. CMS-causing genes have been proposed to activate MRS to mediate changes in nuclear gene expression [2]. For example, in maize, the CMS-causing gene *orf355* regulates the expression of the nuclear transcription factor *ZmDREB1.7* via an MRS-dependent pathway [17]. Direct evidence of MRS in rice CW-CMS was provided by the *Rf17* gene, which encodes a protein with an acyl carrier protein synthase-like domain. A single nucleotide substitution in the promoter region of *Rf17* led to reduced expression in the CMS background, paradoxically restoring fertility [6,7]. In many CMS lines, increased expression of mitochondrial stress marker genes, such as *AOX1a*, has been used as indirect evidence of MRS activation. *AOX1a* upregulation has been reported in maize CMS-S and rice CW-CMS lines [6,17]. In contrast, our RNA-Seq data revealed no change in *AOX1a* expression between Micro-Tom and Dwarf CMS[P] pollen, which is consistent with the unaltered ATP and ROS levels in the CMS line. Despite the lack of ATP or ROS fluctuations, we observed upregulation of four nuclear-encoded *PMEI* genes in CMS pollen. This suggests that *orf137* may induce an alternative MRS pathway that is independent of ATP or ROS. Other known inducers of MRS include the NADH/NAD + ratio, intracellular pH, free Ca^2+^ concentration, and changes in the mitochondrial membrane potential [16]. In wheat CMS, which carries the cytoplasm of *Aegilops crassa*, the CMS-causing gene *orf260* has been suggested to cause the homeotic transformation of stamens into pistil-like organs via MRS-mediated alteration of *B-class MADS-box* genes. In this CMS line, *Wheat Calmodulin-Binding Protein 1* (*WCBP1*) has been proposed as a component of a Ca^2+^-dependent signaling pathway, suggesting that *orf260* may initiate MRS through Ca^2+^ signaling [22,23]. Considering the possible involvement of MRS through Ca^2+^ signaling in the tomato CMS line, we performed gene ontology (GO) analysis of the 440 identified DEGs. This analysis revealed a significant enrichment of the term *calcium ion binding* (GO:0005509), which included nine genes (Appendix A). Among them, five genes (*Solyc01g006730*, *Solyc02g063340*, *Solyc02g091480*, *Solyc06g150132*, and *Solyc10g077010*) were predominantly expressed in mature anthers and pollen, suggesting a strong association with pollen germination (Appendix A). These findings suggest that *orf137* in tomato may similarly activate an unconventional MRS pathway, potentially involving Ca^2+^ or other signaling molecules, leading to upregulation of *PMEI* genes. Future studies focusing on the dynamics of signaling pathways beyond ATP and ROS in CMS pollen are essential to elucidate the molecular connections between *orf137* and nuclear gene regulation.

## 4. Materials and Methods

### 4.1. Plant Materials

Tomato CMS line CMS[P] was previously developed through an asymmetric cell fusion, where the tomato cultivar P served as the nuclear donor and the wild potato relative *Solanum acaule* as the cytoplasmic donor, followed by repeated backcrossing using P [24]. The plants were cultivated in greenhouses at the University of Tsukuba. The dwarf CMS[P] was developed from CMS[P] by backcrossing with the tomato dwarf cultivar Micro-Tom (TOMJPF0001) [25] in our previous study [8]. They were cultivated at 23 °C under a 16/8 h light/dark cycle.

### 4.2. Phenotypic Analysis of Pollen

Pollen germination was assessed in a liquid germination medium and on the stigma, according to previously described methods [10]. Briefly, for pollen germination in liquid medium, freshly opened flowers were placed in a 1.5 mL tube containing 1 mL of germination medium (15.1 *w*/*v*% polyethylene glycol 6000, 10 *w*/*v*% sucrose, 1.63 mM H_3_BO_3_, 1.27 mM Ca(NO_3_)_2_, 1 mM MgSO_4_, 1 mM KNO_3_, and 0.1 mM K_2_HPO_4_; pH adjusted to 7.0) and vortexed to release pollen from the anthers. After removing floral debris, the suspension was incubated with gentle rotation at 25 °C. Pollen tube growth was then examined using a BX53 microscope (Olympus Corporation, Tokyo, Japan). The proportion of normally germinated and abnormal pollen was determined from four independent biological replicates, each including more than 300 pollen grains. For germination on the stigma, pistils were collected 24 h after self-pollination and fixed in ethanol:acetic acid 3:1. Fixed samples were treated with 5 M NaOH for 24 h, rinsed three times with water, and stained with 0.001 *w*/*v*% aniline blue in 0.1 M K_2_HPO_4_ buffer (pH 10) for 24 h in the dark. Pollen tubes were visualized under UV excitation using a BX53 microscope (Olympus Corporation, Tokyo, Japan), and images were captured with a DP72 digital camera (Olympus Corporation, Tokyo, Japan). For DAPI staining, pollen from the anthesis stage was fixed in ethanol:acetic acid 3:1. The pollen was then stained using DAPI solution, which contained 0.1 M phosphate buffer (pH 7.0), 0.5 M EDTA, Triton X-100, and 3.0 µg/mL DAPI, for 5 min and observed under ultraviolet light using an OLYMPUS BX53 (Olympus Corporation, Tokyo, Japan) and ZEISS Axio Observer microscopes (Carl Zeiss, Oberkochen, Germany). For I_2_-KI staining, pollen was collected from anthers at the anthesis stage, and it was stained with a 2 *w*/*v*% I_2_-KI solution. More than 300 pollen grains were observed under the light microscope in each experiment to calculate the staining ratio. ROS levels in the pollen were measured as previously described [26]. The fluorescence intensity of individual pollen grains was quantified using the ImageJ software (version 1.51) [27]. For each experiment, more than 100 pollen grains were analyzed, and the average value was calculated.

The pollen structure was examined using TEM. Pollen grains were incubated in a liquid germination medium for 1 h and then fixed using a two-step fixation protocol. First, samples were treated with 4% paraformaldehyde containing 10% sucrose and 15.1% polyethylene glycol 6000 in 0.05 M cacodylate buffer (pH 7.4) at 4 °C for 1 h. This was followed by a second fixation using 2% paraformaldehyde and 2% glutaraldehyde in the same buffer at 4 °C for 16 h. After three rinses with cacodylate buffer, the pollen was post-fixed with 2% osmium tetroxide in cacodylate buffer overnight at 4 °C. Dehydration was achieved using a graded ethanol series (50%, 70%, 90%, and 100% ethanol) at 25 °C. To ensure complete dehydration, the samples were kept in 100% ethanol for two days. Subsequently, the samples were infiltrated with propylene oxide and a 1:1 mixture of propylene oxide and epoxy resin (Quetol-651; Nisshin, Tokyo, Japan) for 6 h and finally embedded in pure resin overnight. Polymerization was performed at 60 °C for 48 h. Ultrathin sections (80 nm) were prepared using an ultramicrotome (Leica Ultracut UCT, Wetzlar, Germany) and were stained with 2% uranyl acetate and lead citrate. The sections were examined under a TEM (JEM-1400 Plus, JEOL, Tokyo, Japan) at 100 kV, and images were captured using a CCD camera (EM-14830RUBY2, JEOL, Tokyo, Japan).

### 4.3. Ca^2+^ Imaging in Pollen

To create a plasmid that expressed the Ca^2+^ probe in pollen, the *G-CaMP5* sequence was amplified from pCMV-GCaMP5G (Addgene #31788) using the following primers (5′- GGTTTAGTGAACCGTCAGA-3′ and 5′- AGGAGAGTTGTTGATTCACTTCGCTGTCATCATTTG-3′). The *Lat52* promoter fragment was amplified from the genomic DNA of Micro-Tom using primers (5′- AGACCAAAGGGCAATATACTCGACTCAGAAGGTATTG-3′ and 5′- ACGGTTCACTAAACCTTTAAATTGGAATTTTTTTTTTTGGTGT-3′). The vector backbone was amplified from the pBI121 plasmid (Clontech) using primers (5′- ATCAACAACTCTCCTGGC-3′ and 5′- ATTGCCCTTTGGTCTTCT-3′). These fragments were ligated using the In-Fusion HD Cloning Kit (Takara Bio) to construct the *Lat52::G-CaMP5* vector. The resulting vector was introduced into *Agrobacterium tumefaciens* strain GV3101. Transformation of the tomato cultivar Micro-Tom was performed according to the previously described method [28], and it was cross-pollinated with Dwarf CMS[P].

Pollen was collected from plants expressing *Lat52: G-CaMP5* and placed on a solid germination medium (12% *w*/*v* sucrose, 1.2 mM H_3_BO_3_, 1.6 mM Ca(NO_3_)_2_, 1 mM MgSO_4_, 0.1 mM K_2_HPO_4_, 0.5 *w*/*v*% agarose). The solid medium was inverted in a glass-bottom dish for Ca^2+^ imaging, which was performed using a confocal laser scanning microscope (LSM700; Carl Zeiss, Oberkochen, Germany). The G-CaMP5 signal was detected by excitation at 488 nm, and detection was performed between 490 and 635 nm.

### 4.4. RNA Expression Analyses

Total RNA was extracted from pollen that had been incubated for 10 min in a liquid germination medium using the RNeasy Plant Mini Kit (Qiagen, Hilden, Germany). The extracted RNA was treated with RNase-free DNase (Qiagen, Hilden, Germany) and used for the preparation of a sequence library using the NEBNext^®^ Ultra™ Directional RNA Library Prep Kit for Illumina, with poly(A) selection and strand-specific protocol. Sequencing was performed on a NovaSeq 6000 platform (Illumina, San Diego, CA, USA) to produce 150 bp paired-end reads, generating approximately 2 Gb and 13.3 million reads per sample. In addition, RNA-seq data from various tomato tissues [29], as detailed in Appendix A, were retrieved using Fasterq-dump “https://github.com/ncbi/sra-tools/wiki/01.-Downloading-SRA-Toolkit (accessed on 2 April 2025)”.

Trim_galore “https://github.com/FelixKrueger/TrimGalore (accessed on 2 April 2025)” was used to trim adaptors and low-quality reads, with option q 30. Read mapping was performed using HISAT2 [30] with the nuclear genome of the tomato cultivar Heinz 1706 (SL4.0) [31] as the reference genome. Transcript quantification was performed using TPMCalculator [32]. The resulting count data were uploaded to the integrated differential expression and pathway analysis (iDEP2.0) [33] for normalization and identification of differentially expressed genes (DEGs) using DESeq2 [34]. Genes with |log_2_ Fold Change| > 1 and FDR < 0.1 were considered significantly differentially expressed. To comprehensively identify *PME* and *PMEI* genes in tomatoes, HMMER searches [15] were performed using the Pfam domain profiles Pfam01095 and Pfam04043, which correspond to *PME* and *PMEI*, respectively. GO analysis of DEGs was performed using Melonet-DB [35,36] with the tomato cultivar Heinz 1706 genome reference (SL4.0) as the sequence database.

### 4.5. PME Activity

PME activity was measured as previously described [37]. Fifty anthers at anthesis were collected and vortexed in 5 mL of liquid pollen germination medium to release pollen. The suspension was filtered through a 30 µm filter (CellTrics; Sysmex, Kobe, Tokyo) and rotated at 23 °C for 10 min. Pollen was pelleted by centrifugation at 1000× *g* for 1 min at 4 °C. The resulting pellet was resuspended in 100 µL of protein extraction buffer (1 M NaCl, 200 mM Na_2_HPO_4_, pH 6.2) supplemented with 1 µL of Protease Inhibitor Cocktail (Sigma-Aldrich, St. Louis, MO, USA), and the pollen was disrupted using a plastic pestle. Following centrifugation at 15,000× *g* for 15 min at 4 °C, the supernatant was collected. Protein concentration was measured using the Bradford assay [38], and the concentration was adjusted to 2 µg/µL using the protein extraction buffer. Subsequently, 8 µL of the protein solution was mixed with 72 µL of 20 mM Na_2_HPO_4_ (pH 6.2), and 200 µL of 0.5% pectin (Sigma-Aldrich, St. Louis, MO, USA) was added. The mixture was incubated at 30 °C for 3 h and then heat-inactivated at 100 °C for 10 min. After cooling, 70 µL of the reaction solution was transferred to a 96-well plate, and 30 µL of 0.001 U/µL alcohol oxidase (Sigma-Aldrich, St. Louis, MO, USA) was added to each well. The plate was shaken for 15 min, and then 100 µL of reaction buffer (0.02 M acetylacetone, 2 M ammonium acetate, 0.05 M acetic acid) was added and shaken for 30 s. The plates were incubated at 68 °C for 10 min and then cooled to 4 °C for 10 min. The absorbance at 412 nm was measured using a microplate reader (SpectraMax M2; Molecular Devices, San Jose, CA, USA). A standard curve was prepared using a methanol dilution series (3.5–70 nmol) treated in the same manner.

### 4.6. ATP Quantification Assay in Pollen

ATP quantification was performed as previously described [39]. Anthers from ten flowers at the anthesis stage were collected in a pollen germination medium and vortexed to release the pollen. The suspension was passed through a 30 µm filter (CellTrics; Sysmex, Kobe, Tokyo). A 100 µL aliquot of the pollen suspension was added to a 1.5 mL tube containing 400 µL of 7% perchloric acid and 10 mM EDTA and placed on ice for 15 min. Then, 100 µL of 5 N KOH and 1 M triethanolamine were added, and the mixture was cooled on ice for 10 min. Following centrifugation at 10,000× *g* for 5 min at 4 °C to precipitate the neutralized salts, 10 µL of the supernatant was mixed with 990 µL of 50 mM Tris-HCl buffer (pH 8.0). Then, 100 µL of the diluted supernatant was transferred to a white 96-well plate, and 100 µL of the ENLITEN rLuciferase/Luciferin Reagent (Promega, Madison, WI, USA) was added. The solution was gently mixed by pipetting, and luminescence was measured using a Fluoroskan Ascent FL luminometer (Thermo Fisher Scientific, Waltham, MA, USA) with an integration time of 10 s. A standard curve was prepared using a 10-fold serial dilution of 10^−7^ M ATP standard solution (Promega, Madison, WI, USA) in 50 mM Tris-HCl buffer (pH 8), ranging from 10^−7^ to 10^−11^ M. As a blank control, the pollen germination medium without pollen was subjected to the same ATP extraction procedure described above. The pollen concentration in the suspension was determined using a cell counter (OneCell, Hiroshima, Japan).

## Figures and Tables

**Figure 1 ijms-26-08337-f001:**
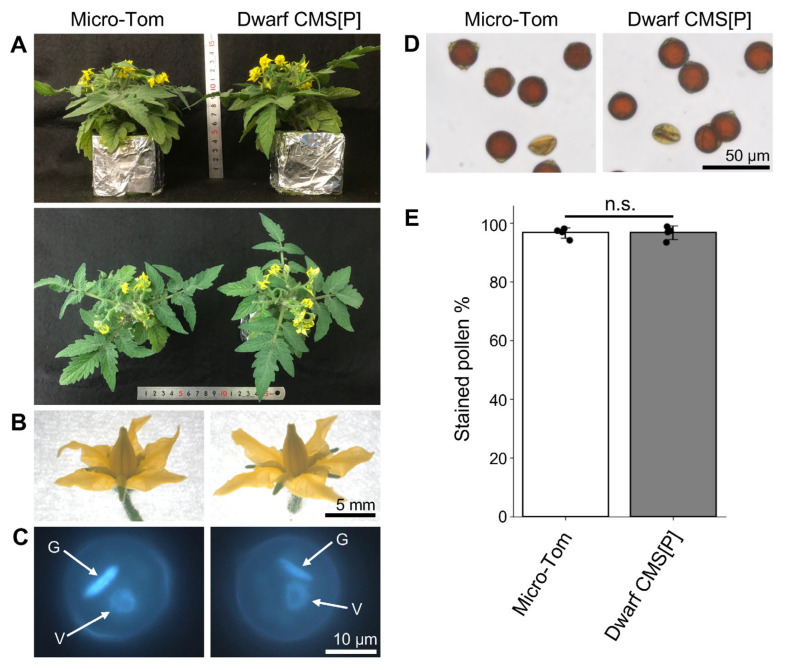
Phenotypic comparison of mature plants, flowers, and pollen grains between Micro-Tom and Dwarf CMS[P]. (**A**) Mature plant morphology of Micro-Tom and Dwarf CMS[P]. (**B**) Flowers of Micro-Tom and Dwarf CMS[P] at the anthesis stage. (**C**) DAPI-stained mature pollen nuclei. V; vegetative nucleus, G; generative nucleus. (**D**) Pollen grains stained with I_2_-KI solution to visualize starch granules. Visible pollen is stained brown. (**E**) Percentage of I_2_-KI solution-stained pollen in Micro-Tom and Dwarf CMS[P]. Data are presented as mean ± standard error from four biological replicates (*n* = 4). Each black dot represents an independent biological replicate. No significant difference (n.s.) was observed between the two lines (Student’s *t*-test).

**Figure 2 ijms-26-08337-f002:**
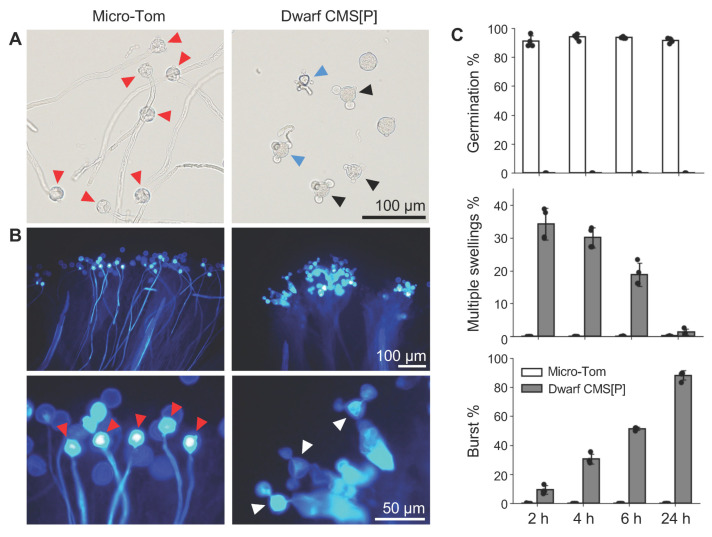
Abnormal pollen phenotypes in Dwarf CMS[P]. (**A**) Pollen phenotypes after 4 h incubation in liquid germination medium. Red arrowheads indicate germinated pollen in Micro-Tom; black and blue arrowheads indicate pollen with multiple swollen apertures and burst apertures, respectively, in Dwarf CMS[P]. (**B**) Morphology of pollen grain and pollen tubes on the stigma 24 h after pollination. Red arrowheads indicate germinated pollen in Micro-Tom; white arrowheads indicate pollen with multiple swollen or burst apertures in Dwarf CMS[P]. Pollen grains and pollen tubes were stained with aniline blue. (**C**) Ratio of germinated pollen, pollen with multiple swollen apertures, and pollen with burst apertures. Data were obtained from four independent experiments (*n* = 4), and more than 300 pollen were counted in each experiment. Each black dot represents an independent biological replicate.

**Figure 3 ijms-26-08337-f003:**
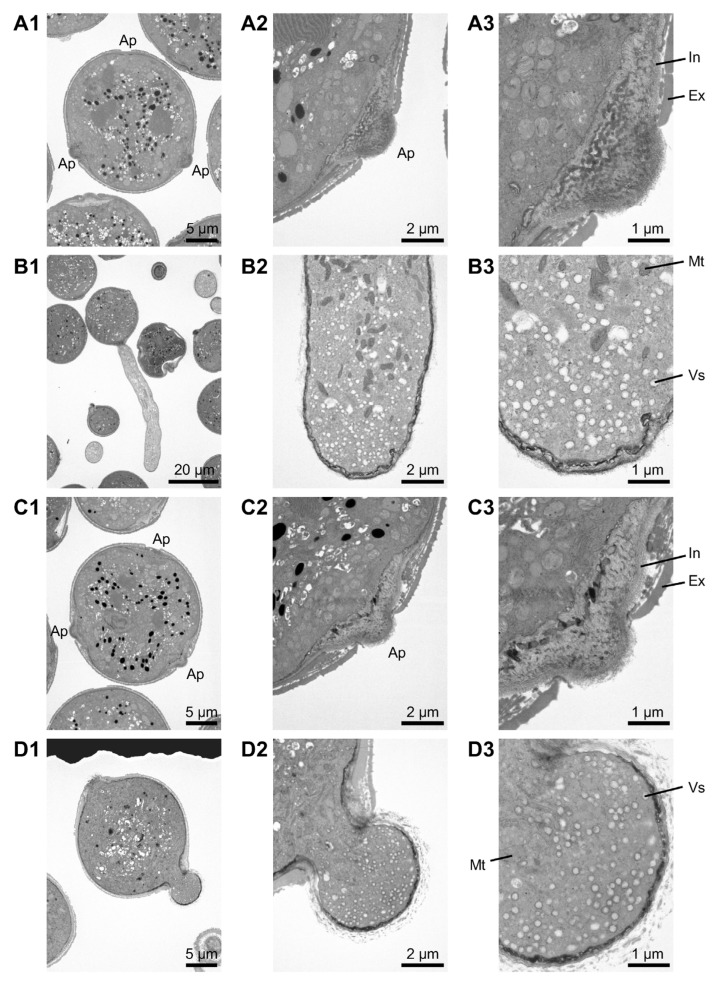
Ultrastructural analysis of pollen before and after incubation by transmission electron microscopy (TEM). (**A1**–**A3**) Micro-Tom pollen before incubation: (**A1**) whole pollen grains, (**A2**) region near the aperture, and (**A3**) higher magnification of the aperture. (**B1**–**B3**) Micro-Tom pollen after incubation: (**B1**) germinated pollen grain, (**B2**) tip region of the pollen tube, and (**B3**) higher magnification. (**C1**–**C3**) Dwarf CMS[P] pollen before incubation: (**C1**) whole pollen grains, (**C2**) region near the aperture, and (**C3**) higher magnification. (**D1**–**D3**) Dwarf CMS[P] pollen after incubation: (**D1**) pollen grain with swollen aperture, (**D2**) tip of the swollen aperture, and (**D3**) higher magnification. Ap, aperture; In, intine; Ex, exine; Vs, vesicle; Mt, mitochondria.

**Figure 4 ijms-26-08337-f004:**
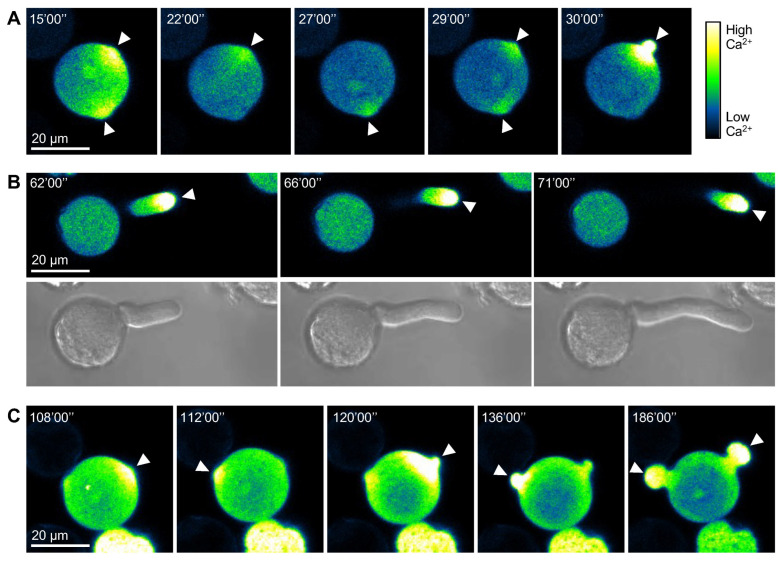
Time-course Ca^2+^ imaging of pollen germination and tube growth in Micro-Tom and aperture swelling in Dwarf CMS[P]. (**A**) Time-lapse Ca^2+^ dynamics during pollen germination in Micro-Tom. (**B**) Time-lapse Ca^2+^ dynamics during pollen tube elongation in Micro-Tom (top: G-CaMP5 signal; bottom: bright-field image). (**C**) Time-lapse Ca^2+^ dynamics in Dwarf CMS[P] pollen showing aperture swelling. White arrowheads indicate regions with elevated Ca^2+^ signals.

**Figure 5 ijms-26-08337-f005:**
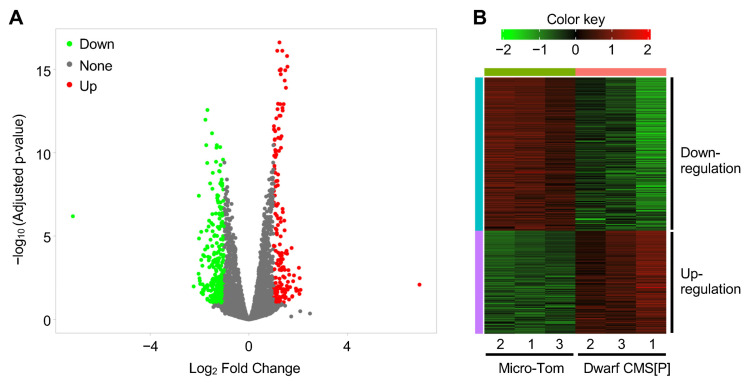
Transcriptome analysis of pollen from Micro-Tom and Dwarf CMS[P] lines. (**A**) Volcano plot of differentially expressed genes (DEGs) identified between Micro-Tom and Dwarf CMS[P] pollen. Genes with |log_2_ fold change| > 1 and false discovery rate (FDR) < 0.1 were considered DEGs. Red and green dots indicate significantly upregulated and downregulated genes in CMS[P] pollen, respectively, while gray dots represent non-significant genes. (**B**) Hierarchical heatmap clustering of DEGs between Micro-Tom and Dwarf CMS[P]. Transcript per million (TPM) values were normalized using Z-score scaling. Each column represents a biological replicate (*n* = 3).

**Figure 6 ijms-26-08337-f006:**
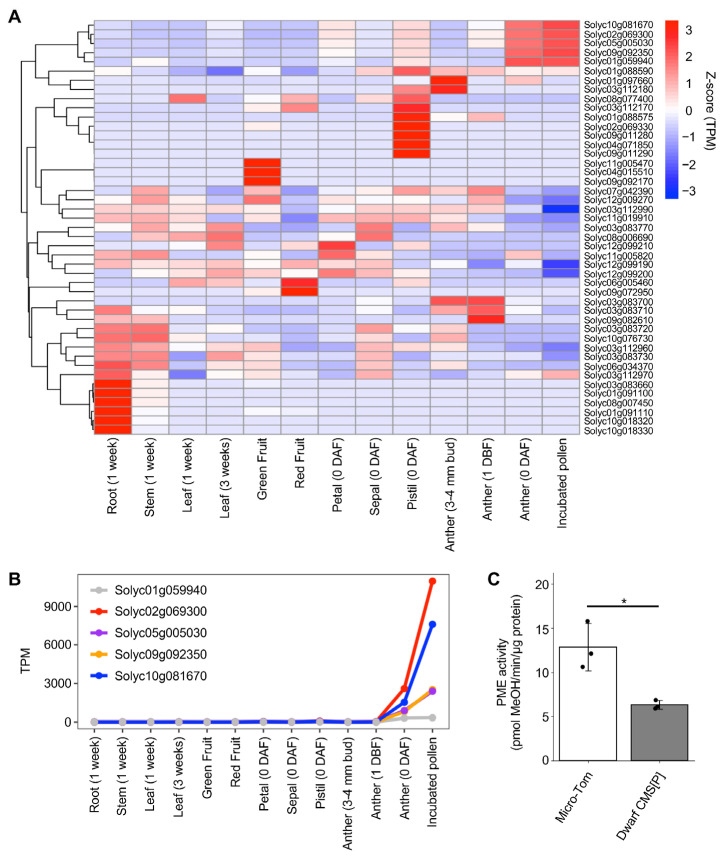
Expression profiles of *pectin methylesterase* (*PME*) and *PME inhibitor* (*PMEI*) genes and their activity in tomato male organs. (**A**) Heatmap showing tissue-specific expression patterns of 45 *PMEI* genes across various tomato tissues. Transcript per million (TPM) values were normalized using the Z-score. A dendrogram on the left indicates hierarchical clustering of genes based on their expression profiles. Three *PMEI* genes (*Solyc08g016100*, *Solyc08g016110*, and *Solyc08g016120*) were excluded from the analysis because their TPM values were zero in all tissues. (**B**) Line graph of TPM values for five highly expressed *PMEI* genes in male reproductive organs across different tissues. (**C**) PME enzymatic activity in the pollen of Micro-Tom and Dwarf CMS[P]. Error bars indicate standard deviation (*n* = 3). Each black dot represents an independent biological replicate. Statistical significance was determined using Student’s *t*-test. * *p* < 0.05.

**Figure 7 ijms-26-08337-f007:**
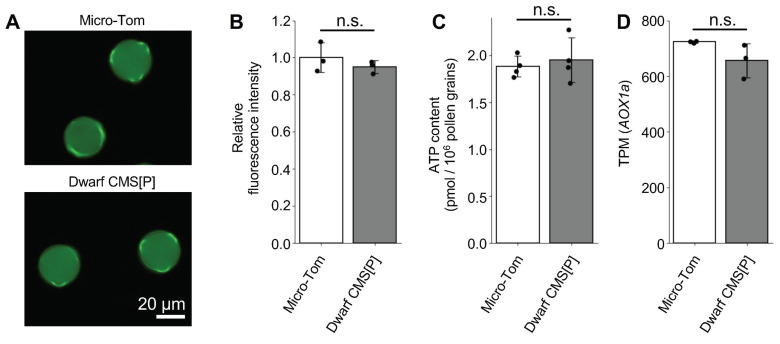
ROS levels, ATP content, and *AOX1a* expression in pollen. (**A**) Representative H_2_ DCFDA-stained fluorescent images of pollen showing reactive oxygen species (ROS) in Micro-Tom and Dwarf CMS[P]. (**B**) Quantification of fluorescence intensity in (**A**). Data are shown as mean ± standard error from three independent biological replicates (*n* = 3). More than 100 pollen grains were measured for each replicate. The average value for Micro-Tom was set to 1. (**C**) ATP content in pollen, measured as pmol per 10^6^ pollen grains. *n* = 3. (**D**) Expression levels of the *AOX1a* gene in pollen as transcripts per million (TPM) values. In (**B**–**D**), each black dot represents an independent biological replicate. No statistically significant difference (n.s.) was observed between Micro-Tom and Dwarf CMS[P].

## Data Availability

The raw RNA-Seq sequences are available at the Sequence Read Archive (SRA) under the BioPro-ject ID PRJNA1250823.

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
