# Peer review of "Characterization and Transcriptome Analysis Reveal Abnormal Pollen Germination in Cytoplasmic Male Sterile Tomato"

_ijms, 2025, doi:10.3390/ijms26178337_

Round 1
Reviewer 1 Report
Comments and Suggestions for Authors
Your study provides valuable insights into the role of Ca²⁺ and the PMEI genes in male sterility induced by the tomato ORF137 gene. The observed differences in Ca²⁺ concentration and transcriptomic profiles significantly contribute to our understanding of the underlying molecular mechanisms. However, there are at least three problems that should be revised.
Firstly, The data of pollen viability and pectin methyl-esterification levels should be supplemented. This would help clarify whether abnormal germination is linked to the PME/PMEI activity.
Secondly, Figure 4 clearly shows the differences of Ca²⁺ concentration. Could the transcriptomic changes reflect the alterations of Ca²⁺-related signaling pathways? Additional discussion on this point would be insightful.
Thirdly, Since CMS[P] pollen exhibited delayed germination compared to that of fertile plants, performing transcriptome analysis after only 10 minutes of culture might not capture key regulatory changes. Could this early time point affect the interpretation of the results?
I appreciate your careful consideration of these points. Please address them in your revision, and I look forward to receiving your revised manuscript.
Author Response
We are very grateful to the reviewer for the insightful and constructive comments. We have addressed all the comments.
Comment 1: Firstly, The data of pollen viability and pectin methyl-esterification levels should be supplemented. This would help clarify whether abnormal germination is linked to the PME/PMEI activity.
Response 1: We agree that data on pollen viability and pectin methyl-esterification levels are important to discuss the male sterile mechanism in tomato CMS. Regarding pollen viability, we have already provided both photographic and quantitative evidence in Figure 1C, D, and E. In this study, we demonstrated that the upregulation of PMEI genes is associated with the abnormal pollen phenotype observed in tomato CMS. Supporting this conclusion, we further showed that PME activity was significantly reduced in CMS pollen, providing consistent evidence that elevated PMEI expression suppresses PME activity.
On the other hand, the precise relationship between the four upregulated tomato PMEI genes, their target PME genes, and pollen germination remains to be clarified. We believe this will require generating individual overexpression and knockout mutants and examining their phenotypes in detail. In this context, measuring pectin methyl-esterification levels will also be essential, and we plan to investigate these aspects in future studies to elucidate the specific roles of individual PMEI and PME genes in linking pectin modification to pollen germination.
Comment 2: Secondly, Figure 4 clearly shows the differences of Ca²⁺ concentration. Could the transcriptomic changes reflect the alterations of Ca²⁺-related signaling pathways? Additional discussion on this point would be insightful.
Response 2: We additionally performed Gene Ontology analysis of the 440 differentially expressed genes and detected a significant enrichment of the term calcium ion binding (GO:0005509). We added this description in the main text, revised Table S1, included a new Table S3, and updated the Materials and Methods section accordingly.
Comment 3: Thirdly, Since CMS[P] pollen exhibited delayed germination compared to that of fertile plants, performing transcriptome analysis after only 10 minutes of culture might not capture key regulatory changes. Could this early time point affect the interpretation of the results?
Response 3: We think that comparing pollen samples with clearly different morphologies, such as germinated versus ungerminated pollen, could lead to the detection of many transcriptional differences that are not directly related to pollen germination. To address this, in our study we compared the transcriptomes of Micro-Tom and Dwarf CMS[P] pollen at the same ungerminated stage (10-min incubation), which allows a more direct exploration of the factors responsible for inhibiting pollen germination in Dwarf CMS[P].
We added an additional explanation in the main text. (Results 2.3)
Reviewer 2 Report
Comments and Suggestions for Authors
The work of Kosuke Kuwabara and Tohru Ariizumi is devoted to studying the current topic of cytoplasmic male sterility. The introduction is clear, detailed, has no unnecessary words and perfectly reflects the essence and purpose of the work, and introduces the reader to the subject.
In the work itself, the authors use a large number of methods, these are microscopy methods, with several staining options, phenotypic assessment, the electron microscopy method, they conduct an assessment of germination and molecular methods. All methods are relevant, relevant and in demand.
I would like to separately emphasize the construction of research logistics, the authors describe in detail the object of their research and the processes that are at the forefront, using different levels of research and methods from different areas of biology.
Despite all that has been said, several questions about the work have arisen:
- Figure 2 B, the authors indicate that this is the morphology of pollen in the pistil after pollination, but do not indicate that there is not only pollen, but also pollen tubes, and do not note in the description the dye that was used.
- Also, the method of assessing the growth of pollen tubes in the tissues of the stigma of the pistil is not described in the "materials and methods" section, there is only a link to the previous publication, the method, at least briefly, but should be described. It will be enough to note the dye, concentration and a few words about the preparation of the temporary preparation.
- Also, there is only a link to the method of germination of pollen.
- The authors indicate the name of the variety in quotation marks, which is incorrect
The work is presented in a competent, understandable language, I think it will be interesting to many researchers and overall, it makes a good impression.
Author Response
We are most grateful to the reviewer for the helpful comments on the original version of our manuscript. We have addressed all the comments.
Comment 1: Figure 2 B, the authors indicate that this is the morphology of pollen in the pistil after pollination, but do not indicate that there is not only pollen, but also pollen tubes, and do not note in the description the dye that was used.
Response 1: We added the description of the pollen tube and the dye. (Figure 2. Legend)
Comment 2: Also, the method of assessing the growth of pollen tubes in the tissues of the stigma of the pistil is not described in the "materials and methods" section, there is only a link to the previous publication, the method, at least briefly, but should be described. It will be enough to note the dye, concentration and a few words about the preparation of the temporary preparation.
Response 2: We added a concise description of the pollen tube staining method. (Materials and Methods 4.2)
Comment 3: Also, there is only a link to the method of germination of pollen.
Response 3: As with Comment 2, we added a concise description of the method. (Materials and Methods 4.2)
Comment 4: The authors indicate the name of the variety in quotation marks, which is incorrect
Response 4: We removed the unnecessary quotation marks. (throughout the manuscript, Figures and Tables)